# Miniaturised implantable circular polarized antenna with a high ARBW

**Rajiv Kumar Nehra[1], Ajay Dureja[2], Rajkumar Singh Rathore[3]\*, Tiansheng Yang[4], Lu Wang[5]**

**1** Dept. of Electronics and Communication Engineering, Bharati Vidyapeeth's College of Engineering, New Delhi, India, **2** Dept. of Information Technology, Bharati Vidyapeeth's College of Engineering, New Delhi, India, **3** Cardiff School of Technologies, Cardiff Metropolitan University, Llandaff Campus, Cardiff, United Kingdom, **4** University of South Wales, Pontypridd, United Kingdom, **5** Xi'an Jiaotong-Liverpool University Suzhou, Suzhou, China

\* rsrathore@cardiffmet.ac.uk

## Abstract

For biological applications, this communication uses an implanted antenna loaded with metamaterial and a sorting pin. The suggested antenna operates at 2.44 GHz in the ISM band. The first antenna's resonant frequency is lowered from 2.53 GHz to 2.46 GHz by applying a sorting pin. This causes the antenna to become circularly polarized and have an ARBW of 580 MHz (2.15 GHz - 2.73 GHz). Strong CP behavior with an ARBW of 830 MHz from 2.01 GHz to 2.84 GHz in the ISM band is produced by incorporation of an H-shaped metamaterial on the antenna's superstrate. Additionally, the reasonable value of the specific absorption rate improved from 960.5 to 952.1. Highlights of the suggested antenna include its miniature size (10.67 mm$^3$), strong CP properties, the significant value of SAR 952.1 W/KG, and unslotted ground plane to detract from designing labyrinthine backscattering radiation. After building the recommended antenna, experiments are conducted using a skin-mimicking gel solution that approximates the electrical characteristics of human skin tissues at 2.44 GHz. In the ISM band, actual and simulated impedance bandwidths of 90 MHz and 110 MHz are acquired, respectively. Together with parametric analysis, simulation and measurement results are consistent.

## 1. Introduction

The biological and telecommunications sectors have recently worked together to ameliorate patient health. To diagnose glucose levels [1], oxygen levels, cardiac monitoring, and other conditions, biomedical devices are produced and implanted in the human anatomy. The implantable antenna is a necessary part of biomedical equipment to establish wireless communication between external receivers and Implantable Medical devices (IMDs) [2]. The implanted antennas' dimensions are a major challenge to make IMDs small and effective. Although the design of a microwave

**Data availability statement:** Uploaded as supplementary information as pdf file along with revised file.

**Funding:** The author(s) received no specific funding for this work.

**Competing interests:** The authors have declared that no competing interests exist.

antenna operating in free space has strong radiation characteristics, lossy media affects the antenna's radiation properties, particularly inside the environment of the human anatomy [3]. Thus, the design and performance of implantable antennas have attracted researchers to construct highly efficient medical devices. Antenna size, radiation efficiency, low SAR, patient safety, biocompatibility, and circular polarization are highly crucial parameters to develop quality IMDs in the medical health system [4]. The three main parts of the IMDs are the antenna, RF circuitry, and battery. Thus, the antenna is the greatest of the three components. Consequently, the size of the antenna must be decreased. However, the antenna's performance suffers due to its significant size decrease [5–6].

The patient must be completely free during the telemetry sessions rather than adopting a tight posture. Many academics have suggested particular antenna designs with CP characteristics to guarantee such high-quality communication between IMDs and external receiving stations. Multipath distortions can be significantly reduced by using CP properties of the implanted antennas. Researchers have investigated a number of methods to achieve the antenna's tiny size and CP characteristic. To construct antennas with CP characteristics and compactness, new methods include sorting pins [7], ground slots [8], and metamaterial [9–10]. Medical equipment that take patient convenience and low multipath path losses into account, are advised to use circular polarization. To achieve this, authors have been working hard to enhance their CP traits. In 2018, Yudi Zhang and colleagues demonstrated a broadband antenna of a circular polarization [11]. The proposed antenna is converted to a circularly polarized antenna by sandwiching the ground plane and radiating planes with a sorting pin. By optimizing sorting pin locations or by using U-shaped slots, 19.7% of 3 dB bandwidth was received in low ISM band. U-shaped slots and sorting pins were added after L-shaped slots were installed to lengthen the current trajectory. An excellent broadband CP characteristic has been acquired in the intended band. In their paper [12], Li Jie Xu et al. introduced implantable antenna with a volume of 91.9 mm3. On the patch surface, cross slots—a conventional method—have been incorporated. The arc-crossed slots span the 2.18 GHz to 2.62 GHz frequency range with an axial ratio bandwidth of 18.3%. Abdenasser Lamkaddem et al. [13] introduced an implantable antenna in 2022 with compact size. The author of this study did not employ intricate methods such as sorting pins. The U-shaped slot structure and meander-shaped antennas have been introduced. The antenna's electrical dimension has altered, but mechanical dimension has remained unchanged. 17.2% axial ratio bandwidth has been reached between 850 MHz and 1010 MHz using this U-shaped slot. The small and low CP characteristics of all the newly described implantable antennas have been studied. The suggested research project has been offered with an incredibly huge axial ratio bandwidth of 41.29% and a highly miniaturized size of 10.67 mm3 with technologies like sorting pin and metamaterials after a thorough review of the current literature on CP implantable antennas. MIMO implantable antennas with dual ISM bands and bandwidths of 23.6% and 12.14%, respectively, were presented by the authors in the publication [14]. In contrast to this suggested

study project, the intended antenna is built with a big dimension of 28.81 mm³. An outstanding link margin of 10 dB over an 8-meter distance is addressed for an implantable antenna with a modest volume of 7.17 mm³ [15]. Significant SAR standards are used to accomplish the high data rates, but the suggested antenna's greater axial ratio bandwidth and circular polarization are its strongest points. The suggested research project is conducted with about the same antenna volume as [15], and it has superior circular polarization properties and bandwidth. With a highly compact volume of 9.44 mm³, a massive bandwidth of 3.39 GHz spanning numerous frequency bands is achieved [16]. The authors of this research have obtained acceptable SAR values at a very high data rate. This publication has had a lot of bandwidth lately, yet it was unable to improve its CP characteristics. Overall, the proposed study effort discusses and achieves every aspect of implanted antenna performance. This paper presents the construction of a triband antenna that can be electronically tuned to operate at several frequencies [17]. The proposed antenna is designed to operate in sub-6GHz bands at 2.45 GHz (ISM, Wi-Fi, and WLAN), 3.3, 3.5, and 3.9 GHz (WiMAX), and 4.1 & 4.9 GHz (4G & 5G). This is achieved by attaching two open-ended stubs to a modified triangular patch radiator using PIN diodes. However, the high miniatured size and CP characteristics were not covered by the suggested antenna. Additionally, the fabrication appears complicated due to the integration of passive components. This study designs and improves the isolation of a filtering MIMO antenna [18] with a radiation null for out-of-band suppressions suitable for 5G sub-6 GHz communications. The MIMO antenna offers -10 dB impedance bandwidth functionality at the most prominent portion spectrum of the 5G NR n78 band, which extends from 3.4 GHz to 3.61 GHz, to enable wireless applications in base stations. The ISM frequency bands standard, human environment phantom, and low losses of multipath fading mechanism, however, do not accept the suggested antenna. This study suggests a simple, compact, dual-band approach that uses the 2.4/5.8 GHz spectrum to satisfy the demands of modern wearables and communication systems [19]. To remove the antenna from a circular monopole, a number of patches and stubs are installed. Both in lower and higher ISM bands, the authors were able to achieve good directional gain. However, body analysis is the focus of the entire research project, which reduces the likelihood of performance accuracy in comparison to body analysis. Furthermore, because the proposed antenna lacks CP characteristics, it experiences multipath fading, which results in significant signal losses during telemetry transmission.

The authors of this study have optimized a spiral patch in a 10.67 mm³ volume. A sorting pin with a radius of 0.2 mm is positioned via the substrate between the patch and ground surface to create an antenna in standard frequency bands (ISM). This sorting pin lowers the antenna's resonance frequency between 2.53 and 2.46 GHz. Furthermore, implantable antennas have been found to have good CP properties. An H-shaped metamaterial is applied to the superstrate surface to further strengthen ARBW. An ARBW enhancement of 830 MHz has been practised along with an acceptable value of SAR 952.1 W/KG for patient safety. Numerous human bodily tissues, including skin, fat, and muscle models, have been thoroughly investigated using the suggested implantable antenna. A thorough investigation is also conducted into the impact of the suggested implantable antenna's skin depth penetration. Various dielectric constants (substrate materials) are also considered for parametric study of the proposed antenna. Different distances between IMDs and external receivers are used to represent the connection margin computation in detail. With a high axial ratio bandwidth of 830 MHz and 10.67 mm³ size, the suggested implantable antenna has excellent CP characteristics.

The Ansys HFSS program is used to run the full simulation. We created the suggested implanted antenna and successfully tested it in a microwave lab's saline solution to confirm the simulation results. Both outcomes are well compared in every simulation and measurement scenario.

## 2. Designing sequence, optimization, and fabrication of proposed antenna

The authors have covered all the specifics of the antenna's final design and optimization processes, including final fabrication, in this section.

### a) Skin Tissue Model for Simulation

Generally, the simulation, parametric analysis, measurement and validation purpose of all implantable antennas are carried out in two different models given as:

1. Vivo Model

2. Vitro Model

Implantable antennas are evaluated inside bodily organs using an in vivo model. Although this method is highly accurate, it needs a license approval from the Government Medical Authority, which is not ideal for educational institutions. An additional technique is the in vitro model, which simulates and tests implantable antennas outside of the body. At a certain frequency, a gel with characteristics similar to those of skin or other human body tissues is created in an in vitro model. Because of its similar electrical characteristics, the skin tissue replica is used as a skin phantom in this study using the in vitro model approach.

A single-layer skin phantom with a proportion of 100 mm each side of radiation box is assigned. The suggested antenna is situated at 4.25 mm depth from the skin phantom top surface shown in Fig 1. The electrical parameter of skin phantom alters based on the operating frequency. At 2.45 GHz, the skin phantom's relative permittivity ($\varepsilon r$) is 38.1, and its bulk conductivity ($\sigma$) is 1.44 (S/m). A full-wave electromagnetic simulation tool on HFSS software is used to simulate and implement the proposed antenna inside the skin phantom using the finite element method.

### b) Designing Steps of Proposed Implantable Antenna

Considering the human body, antennas must be as small as possible. Typically, a substrate of high relative permittivity is chosen to provide compactness and superior antenna performance, including radiation pattern and input reflection coefficient. The same dielectric substance is also employed as a superstrate to keep antenna patches away from bodily tissues. In this study, a spiral-shaped implantable antenna is proposed.

The specific side proportion of 7 mm x 6 mm of antenna are taken. The Rogers RT Duroid 6010/6010LM material of the antenna has a $e_r$ 10.2 and a thickness of 0.127 mm. A radiating surface with a spiral form aid in extending the antenna's current path. The superstrate material covers the patch surface of the implanted antenna. This material is intended to lessen the dielectric mismatch between the dielectric substrate and the human environment. Consequently, the amount of undesired radiation is reduced due to a significant reduction in the linking of the human body and the antenna. It also helps to make implanted antennas biocompatible and mechanically durable. So, superstrate layer is also used on the patch area of an antenna to reduce

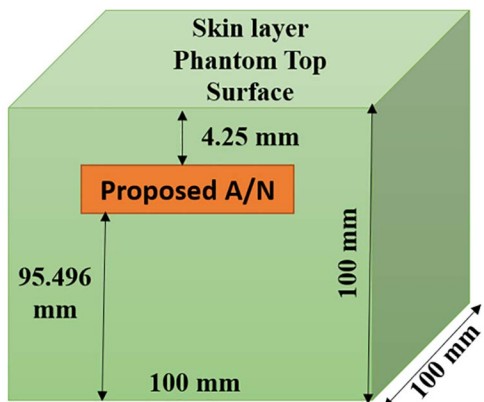

**Fig 1. Configuring the suggested antenna for simulation in a single-layer skin phantom.**

extra radiation in an undesired direction, additionally aid in enhancing the radiation property, and avoid human tissue contact with the patch surface [20]. The thickness of the superstrate is the same as that of the substrate material. This antenna has a very compact structure of a volume of 7mm X 6mm X 0.254 mm (10.67 mm$^3$). The volume of a proposed implantable antenna is probably the least among the latest published research work. The antenna geometry is displayed in Fig 2 below.

   The spiral-shaped antenna is excited by coaxial feed with a specific position (2.25,3.5,0). In coaxial feed, a cylinder's inner and outer radii are selected to correspond with the 50-ohm coaxial connector, respectively. There is a spiral structure on the patch surface. This one has a bigger perimeter than earlier planer structures. Consequently, a larger perimeter results in a larger current conducting path, raising the inductance value. Due to the high inductance, the resonating frequency is significantly decreased. The antenna has a spiral form to lengthen the current path and restrict resonating frequency to its lower value. Generally, Researchers employ to present unequal slots on patches or ground plane which cause disturbance in orthogonal modes and hence build implantable antenna circularly polarised. However, because of

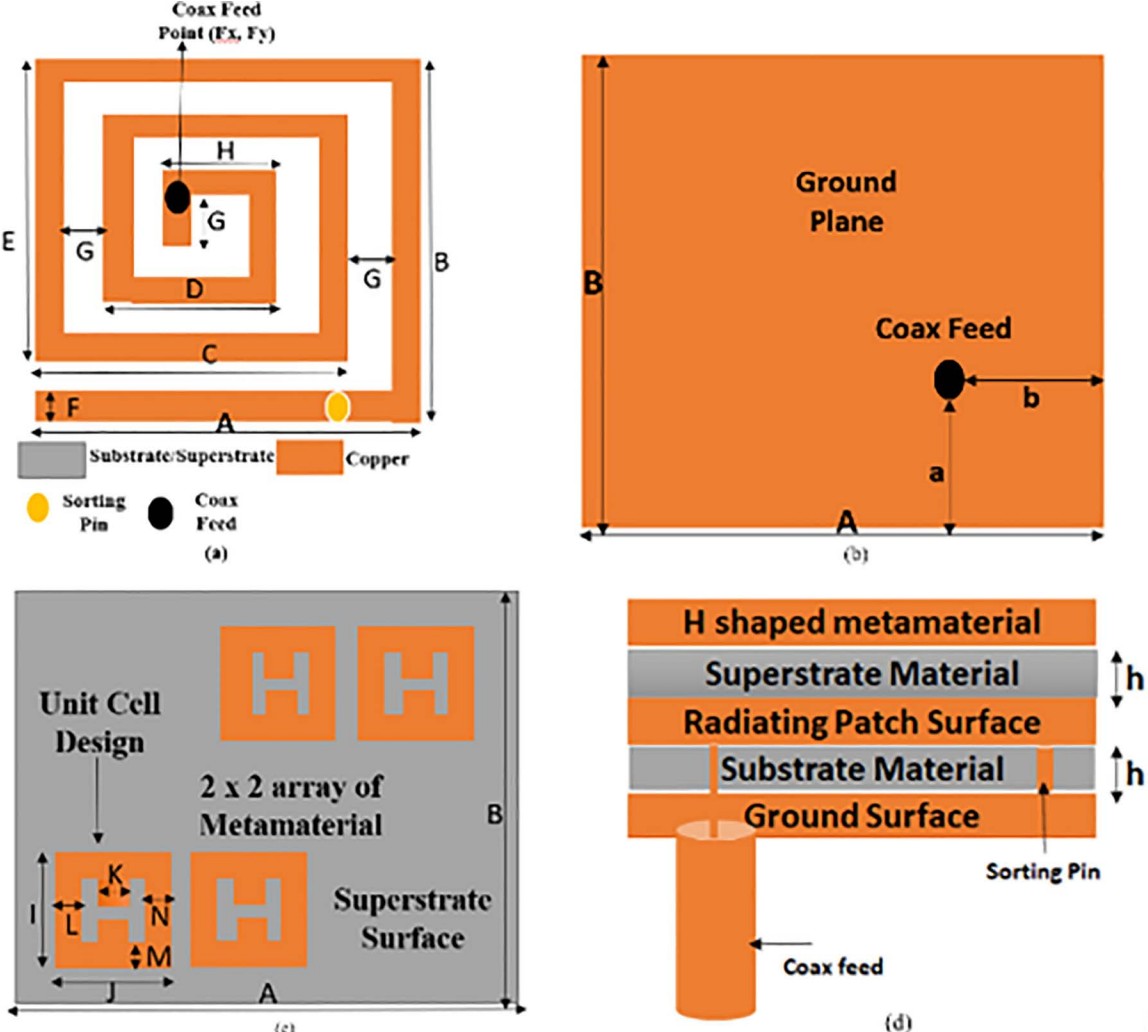

**Fig 2. (A)** Design of patch surface of proposed antenna from top view **(B)** ground plane of proposed antenna from bottom view **(C)** Design of Metamaterial on superstrate **(D)** Side view of proposed antenna.

ground slots, there is always a potential of back radiation, which can result in back lobes and stop serving as a reflector for electromagnetic wave radiation. So, in this research work, a sorting pin is applied between patch and ground plane at a specific position (6,0.25,0) after observing the parametric study on different patch positions. The sorting pin has radius of 0.2 mm which makes a sorted path between the patch and ground of the proposed antenna. In the end, a metamaterial H-shaped structure on the superstrate is used which Fig 2C displays. The antenna design features are presented in Table 1.

## 2.1. The suggested implanted antenna's design process

The experience of the authors with literature surveys indicates that a significantly smaller implanted antenna is required. During antenna design, a large increase in the conducting length on patch is needed to reduce the resonance frequency. Forms that meander and swirl are therefore highly favored. But in the interim, other elements like bandwidth and the axial ratio are degrading. In this investigation, every performance is achieved within the targeted frequency range. The design of the suggested antenna features a spiral construction. Another way to increase the channel conducting is to add a sorting pin or via through patch and ground. The current distribution of antenna is eventually shifted and becomes circularly polarized. The H-shaped metamaterial is implanted on the superstrate surface to improve the CP properties even more. The employment of metamaterial, sorting pin, and spiral processes has enabled circular polarization to be significantly reduced in size and have an exceptionally wide bandwidth.

The dimension of antenna is taken from standard equation of microstrip antenna (MSA). The novelty of proposed antenna design is mentioned as per procedure followed sequentially.

1. Standard rectangular structure is taken with length 7 mm, width 6 mm and thickness 0.127 mm. The rogers rt duroid of dielectric constant of 10.2 is considered. This microstrip antenna is resonated at 6.77 GHz.

2. Then the spiral structure is implemented to increase the current path and eventually spiral path antenna is resonated at 2.53 GHz.

3. After adopting spiral structure, the sorting pin and metamaterial are integrated. The resonance frequency is lowered to 2.44 GHz. High reduction in size has been obtained in final suggested antenna.

In this section, the complete design procedure of the proposed implantable antenna is presented. Three steps are taken into consideration to design the final structure of the proposed antenna. Optimization has been endowed in every design step of the proposed antenna. The various designing steps are shown in Fig 3. The complete study and

**Table 1. Details of proposed implantable antenna parameters.**

| Parameters | Values (mm) | Parameters | Values (mm) |
|---|---|---|---|
| A | 7 | J | 1.275 |
| B | 6 | K | 0.4 |
| C | 6 | L | 0.3 |
| D | 4 | M | 0.1 |
| E | 5 | N | 0.175 |
| F | 0.5 | a | 1.785 |
| G | 0.5 | b | 2.035 |
| H | 3 | h | 0.127 |
| I | 1.025 | (Fx, Fy) position | 2.25, 3.5 |
| (Sx, Sy) position | 6, 0.25 | | |

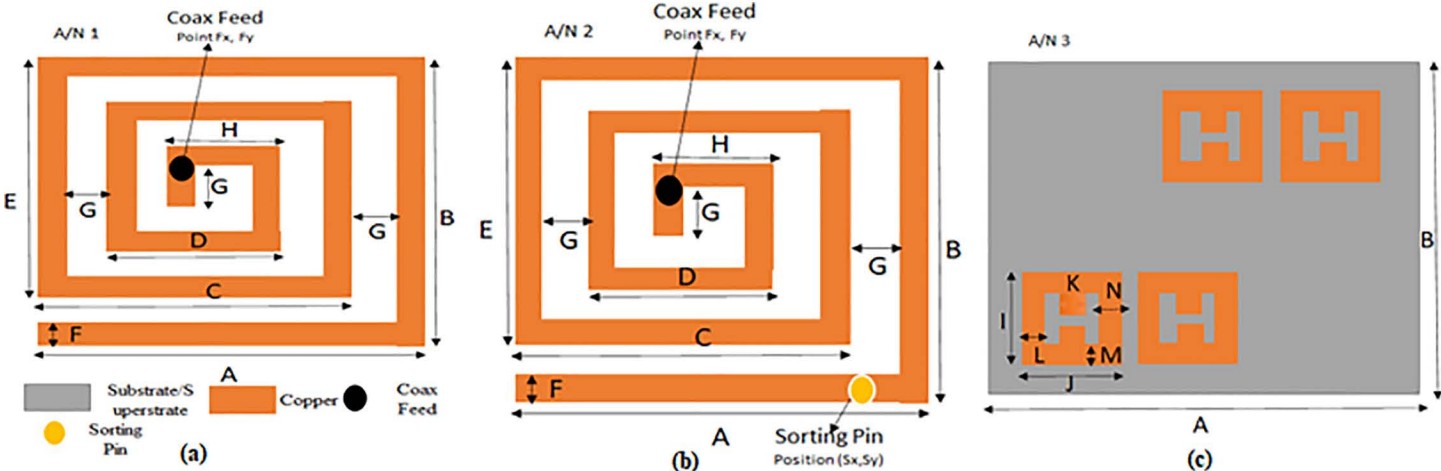

**Fig 3. (A) Design of A/N 1 radiating surface.** (B) Design of A/N 2 radiating surface. (C) Design of metamaterial of A/N 3 on the superstrate surface.

comparison of the reflection coefficient of various designs are presented in Fig 4A. CP behavior of all antenna designs is depicted in Fig 4B.

**2.1.a) Step 1 (A/N 1).** At first, a spiral-shaped design was considered to lengthen the current path on radiating surface which is depicted in Fig 3A. The volume of A/N 1 was taken as 7 mm x 6 mm x 0.254 mm (10.67 mm³). The feed position of coax feed is obtained by optimizing the entire radiating surface of the patch and thus achieving resonating frequency at 2.53 GHz which is near to the ISM band but received poor impedance matching with regard to reflection coefficient due to the lossy environment of the human body shown in Fig 4A. This antenna does not show circular polarization as shown in Fig 4B.

**2.1.b) Step 2 (A/N 2).** A/N 1 is succeeded by inserting a sorting pin through patch and the ground plane. By the introduction of the sorting pin, the current distribution is completely altered. Optimization of sorting pin position worked well at Sx, Sy coordinates. At this specific position of the sorting pin, A/N 2 exhibits resonating frequency at 2.46 GHz (ISM Band) with a reflection coefficient of -30.64 dB shown in Fig 4A. The frequency range for the impedance bandwidth (120 MHz) is 2.4 GHz to 2.52 GHz. With the inclusion of a sorting pin, A/N 2 behooved into circular polarization with ARBW 580 MHz (2.15 GHz - 2.73 GHz) in Fig 4B.

**Parametric analysis of sorting pin at various patch positions of A/N 2.** Spiral-shaped reference antenna (A/N 1) is installed with a sorting pin through patch and ground. The addition of a sorting pin makes this antenna as PIFA which is termed as A/N 2. For the performance analysis purpose, the sorting pin is moved at various patch positions. Fig 4D demonstrates the performance of the A/N 2 w.r.t axial ratio. 8 different positions of the sorting pin are marked with a number from 1 to 8 as shown in Fig 4C. From position 1–7, the axial ratio remains above 3 dB from 2 GHz to 3 GHz which resulted in this antenna into linear polarised. But at position 8, the antenna shows an axial ratio not more than 3 dB which makes this antenna circular polarised. So, position 8 of the sorting pin is prime responsible position for turning the antenna into CP. A good ARBW is achieved ranging 2.15 GHz to 2.73 GHz at this position. Different positions are represented in Table 2 given below.

**2.1.c) Step 3 (A/N 3).** To further improve the CP behavior of A/N 2, an H-shaped metamaterial is imposed on the superstrate surface in diagonal symmetry manner [21]. This metamaterial structure made it the final proposed implantable antenna which is termed A/N 3. With the introduction of metamaterial of H-shaped structure on superstrate, the resonating frequency of A/N 2 is shifted from 2.46 GHz to 2.44 GHz with -22.99 dB reflection coefficient. Impedance bandwidth of 110 MHz (ranging from 2.38 GHz to 2.49 GHz) is achieved as shown in Fig. 4 (a). A great advancement of axial ratio

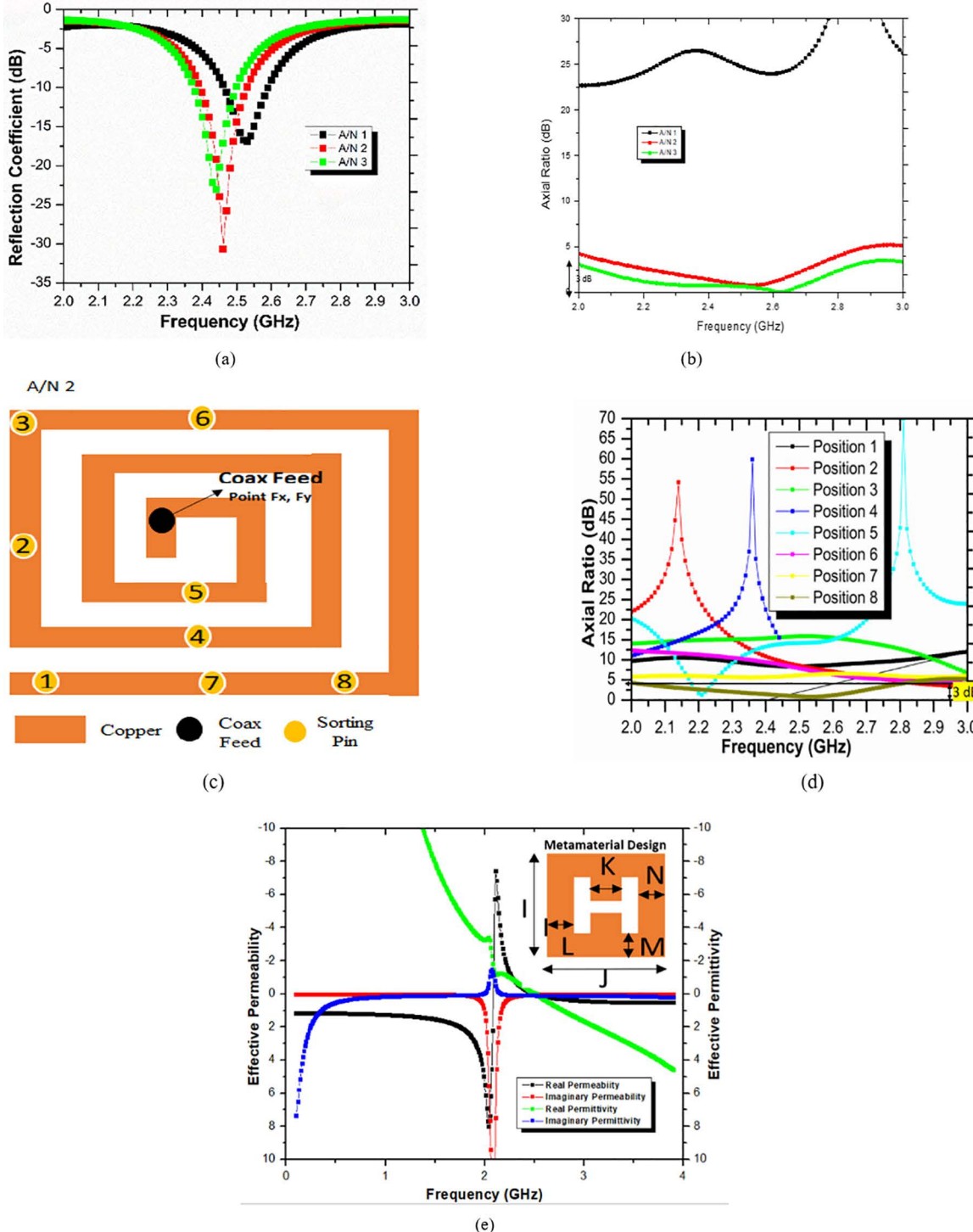

**Fig 4. (A) Reflection coefficient comparison of several antenna designs.** (B) Comparison of axial ratio bandwidth of different antenna designs. (C) Position of sorting pin on patch surface. (D) The axial ratio of different positions of sorting pin on patch surface (A/N 2). (E) Characteristics curve of metamaterial design with effective permittivity and permeability.

**Table 2. Different sorting pin positions on A/N 2 patch surface.**

| Sorting Pin Position Number | Coordinate points (mm) |
|---|---|
| 1 | 0.5, 0.25, 0 |
| 2 | 0.25, 3, 0 |
| 3 | 0.25, 5.75, 0 |
| 4 | 3, 1.25, 0 |
| 5 | 3, 2.25, 0 |
| 6 | 3, 5.75, 0 |
| 7 | 3.25, 0.25, 0 |
| **8** | **6, 0.25, 0** |

bandwidth is instated. It is noted from Fig 4B that the Axial ratio bandwidth of A/N 3 is 830 MHz (2.01 GHz - 2.84 GHz). The material of the high-dielectric superstrate stabilizes the effective permittivity fluctuations surrounding the antenna and decouples it from the lossy environment. According to Lovat et al.'s detailed discussion [22], an MTM superstrate with EVL ($|\varepsilon r| \gg 1$) and mu extremely large ($|\mu r| \gg 1$) characteristics is necessary to increase the magnetic dipole's broadside gain, radiation efficiency, and directivity. Additionally, a wave source excites a grounded MTM structure of permittivity $\varepsilon r$ and permeability $\mu r$ in order to match impedance at resonating frequency and consequently improve overall radiation properties of antenna. The proposed antenna loaded with 2 x 2 array MMT design on superstrate surface shows huge permittivity $\varepsilon r$ and permeability $\mu r$ values which supports the transmission of radiation due to matching with surrounding intrinsic impedance. The characteristics curve of complex permittivity $\varepsilon r$ and permeability $\mu r$ and metamaterial behaviour are well presented in Fig 4E.

## 2.2. Parametric analysis of the suggested implanted antenna in different aspects

The various depths at which various bodily tissues are pierced offer comprehensive information about how well antennas work. Therefore, different tissues and the degree of penetration are simulated before the final suggested antenna is created. In connection with the recommended antenna design, a parametric study of several dielectric materials is also looked at.

**2.2.a) Impact of the skin box's penetration depth.** A skin box that resembles human skin is taken with dimensions 100mm x 100mm x 100mm shown in Fig 1. The proposed antenna is engrafted with different depths of penetration D. The penetration depth D varies from 3mm to 21mm with a 3mm step size. The reflection coefficient of such a parametric study is depicted below in Fig 5A. Fig 5A demonstrates that the proposed antenna does not show a major change in resonating frequency.

**2.2.b) Effect of different human tissues on proposed antenna performance.** Skin, muscle, fat, and bone are among the several human tissues that are taken into account when testing the effectiveness of the suggested antenna. Fig 5B shows the parametric analysis of the reflection coefficient in various tissues. The suggested implantable antenna is designed to penetrate various human body tissues at a depth of 4.25mm. It is investigated that the skin tissue antenna performs well in terms of input reflection coefficient at 2.44 GHz. The suggested antenna changed the resonating frequency from 2.44 GHz to 2.37 GHz with an impedance bandwidth of 120 MHz from 2.31 GHz to 2.43 GHz when muscle tissue is taken into account as a radiation box. If muscle tissue is utilized for implantation, the intended antenna's nearly ISM band is absent. The suggested antenna is totally deteriorated and useless as a radiator in the fat layer. Finally, the suggested antenna is inserted into bone tissue at the same depth of 4.25mm. As of right now, the antenna's 2.75 GHz reflection coefficient is -8.76 dB, which is too low for wireless communication between a biological device and an external receiver. It is clear that the suggested antenna only functions properly in skin tissues and performs exceptionally well in the ISM band. So, proposed antenna is recommended for skin tissues only.

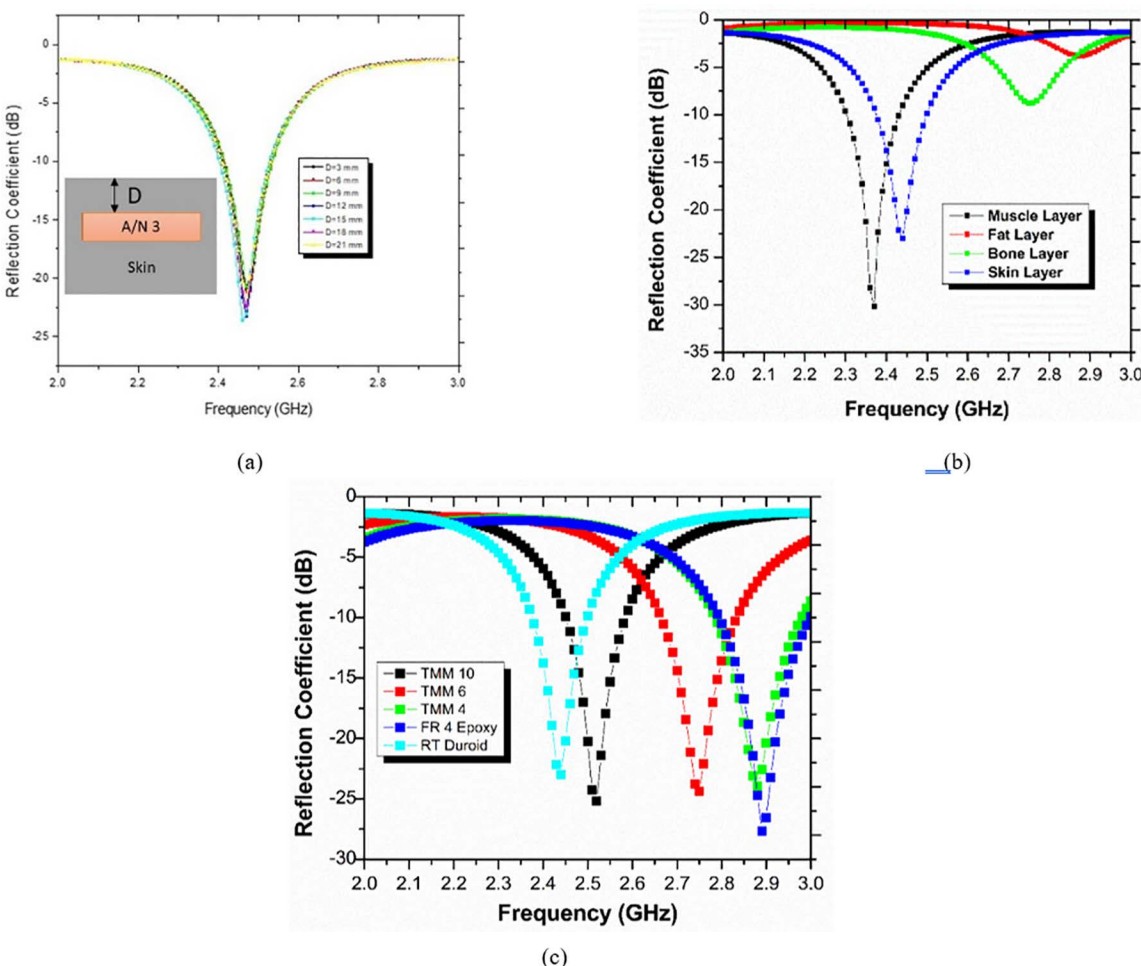

**Fig 5. (A) The suggested antenna's (A/N 3) reflection coefficient at various penetration depths.** (B) The reflection coefficient of the proposed antenna in several human tissue models. (C) Various dielectric substrate materials and the reflection coefficient of the proposed antenna.

**2.2.c) Effect of dielectric material of the proposed antenna.** The substrate of an antenna is a key part to provide physical strength and mechanical support. For the PIFA antenna, researchers use dielectric material as a substrate. Various substrates are investigated in deep. Before finalizing the substrate material for the proposed antenna design, authors explored different substrates with different dielectric constants such as Rogers RT/duroid 6010, TMM4, TMM6, and TMM10 with a thickness of 0.127 mm and dielectric permittivities of 10.2, 4.5, 6.02, 9.8 respectively. All substrate materials are family members of Rogers material. All materials have the same tangent loss with different dielectric constant. Fig 5C depicts the study of reflection coefficient for different substrate materials. It is precisely noted that the proposed antenna design exhibits the best impedance matching in the ISM band with Rogers RT/duroid 6010 material. All other substrate materials have shifted resonating frequency to a higher value which is outside the ISM band. The FR4 epoxy is the least expensive of the materials listed, however it exhibits good impedance matching at 2.89 GHz with a reflection coefficient of -27.65 dB, which is outside the ISM band. The analysis of different materials suggests adopting Rogers RT/duroid as best suited material for this research work.

**2.2.d) Effect of superstrate material and shape of the proposed antenna.** The suggested antenna is thoroughly examined in a variety of biocompatible superstrate materials and forms. Biocompatible superstrate materials such as FR-4 epoxy, polyethylene, alumina, etc. are used to analyze the suggested antenna. Superstrate is accomplished in a variety of shapes, including triangular, cylindrical, and rectangular. Fig 6 below shows the suggested antenna's performance in terms of SAR and reflection coefficient.

Fig 6A makes it evident that the suggested antenna is unsuitable for use in ISM bands, with the exception of Roger R. Duroid biocompatible superstrate material. However, it is unexpected to find that biocompatible materials such FR-4 epoxy, polyethylene, and alumina have significantly improved SAR, as seen in Fig 6B, 6C, and 6D. The SAR value of proposed antenna is greatly improved by opting different shapes of superstrate as shown in Fig 6E. In Fig 6F, it has been stated that except rectangular shape of superstrate all other shapes have shifted resonant frequency out of ISM band. So, proposed antenna is simulated and fabricated with rectangular shape of Rogers rt duroid material.

**2.2.e Proposed antenna with biocompatible layer.** The human body is an excellent heat or radiation absorber, and the implantable antenna is made of conductive material. Therefore, if the implantable antenna's conducting material comes into contact with bodily organs, it could seriously harm those organs. However, implantable antennas typically need to be placed inside biomedical devices. To prevent any negative reactions with human body tissues, all such biomedical equipment are often covered in biocompatible materials such as ceramic alumina, macor, Teflon, FR-4 Epoxy, Roger RT duroid, etc. Thus, the biocompatible material (Rogers RT Duroid) is incorporated into the suggested antenna and provided in all aspects, guaranteeing patient safety as well. Since the suggested antenna's SAR and biocompatible material are within an acceptable range, the patient won't be harmed. The $S_{11}$ of proposed antenna with biocompatible material shows resonating frequency at 2.48 GHz with value of -20.79 dB of impedance bandwidth from 2.42 GHz to 2.53 GHz (110 MHz) and ARBW of 2 GHz to 2.59 GHz as shown in Fig 7. The SAR value of suggested antenna with biocompatible layer is 909.1 W/KG which is acceptable for patient safety.

## 2.3. Antenna fabrication

The proposed implantable antenna (A/N 3) is manufactured with the use of printed circuit board technology. The model implantable antenna with dimension 7 mm X 6 mm X 0.254 mm is manufactured with substrate Roger RT/Duroid 6010/6010LM possessing a loss tangent (σ) of 0.0023 and a dielectric constant εr = 10.2. A cautious effort is taken place to install a coax feed in such a miniaturized size of the proposed antenna. To install sorting pin through the substrate, drill machine of radius 0.2 mm prepared hole. Later, authors inserted copper wire of the same radius through the substrate which simply sorted the ground and patch surface together. Fig 8 demonstrates the proposed configuration of fabricated antenna prototype of radiating patch surface and metamaterial loaded design on superstrate.

## 3. Measurement and discussion

This section covers the measurement setup and a detailed analysis of the fabrication process for the suggested implanted antenna. This section goes into detail on the simulation and the outcomes of the tests. The practical communication between transmitter and receiver is established by calculating link margin analysis.

## 3.1) Antenna performance measurement setup

Simulation work is successfully endowed with finite element method in Ansoft HFSS. The suggested implantable antenna is dipped in skin mimicking gel at 4.25 mm. Skin mimicking gel was replicated by a 3D box in Fig 1 with skin properties at 2.45 GHz in Ansoft HFSS. Skin characteristics such as bulk conductivity and dielectric constant for ISM bands are obtained from [23]. The results of the constructed structure of the suggested implantable antenna are measured using the KEYSIGHT Field Fox RF Analyser N9914A 6.5 GHz network analyzer in Fig 9A in order to verify the performance of the proposed antenna. Since an implantable antenna goes inside a human anatomy, so to emulate the human tissue model

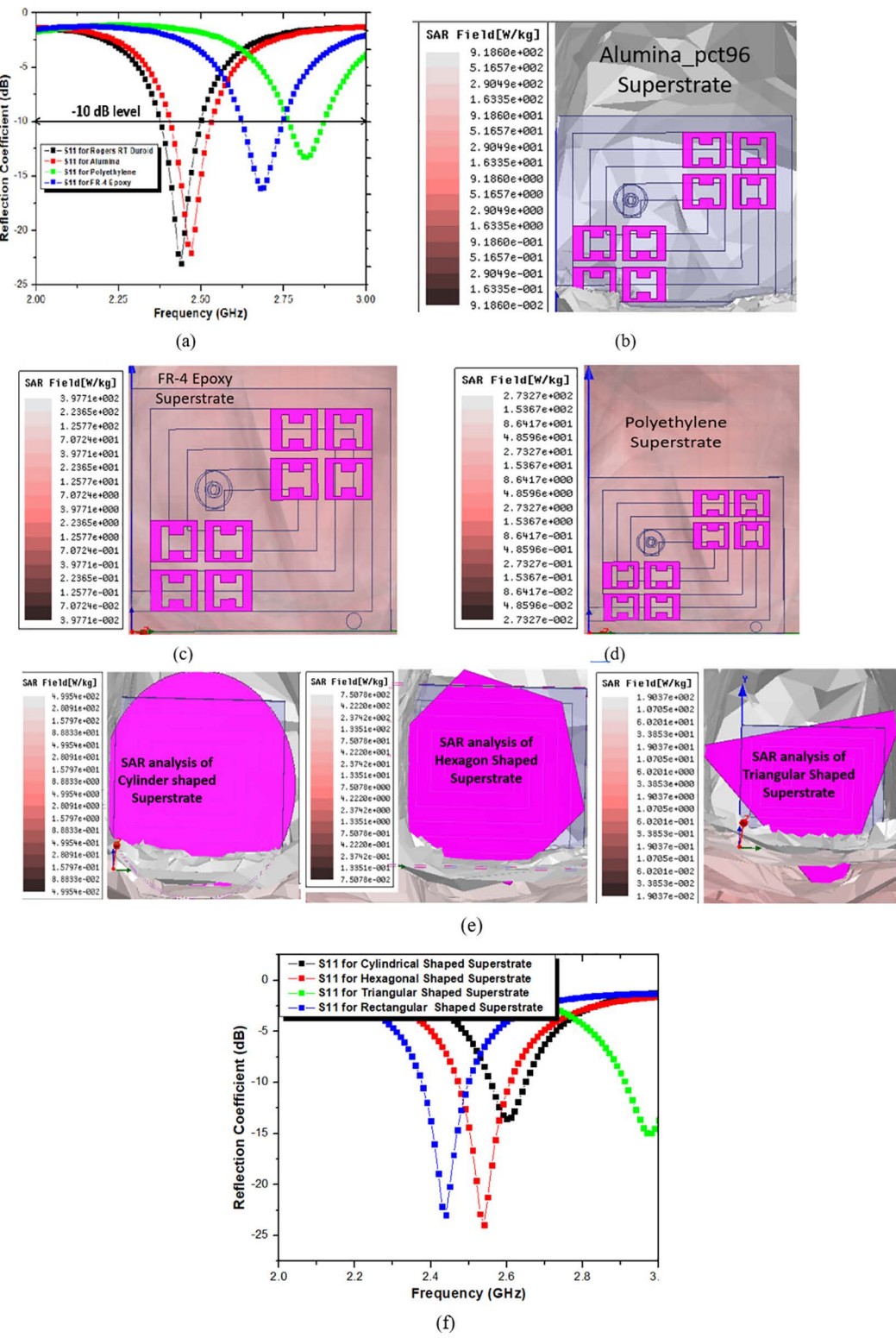

**Fig 6. (A) Reflection Coefficient of different superstrate materials.** (B) SAR of Alumina superstrate material. (C) SAR of FR-4 epoxy superstrate material. (D) SAR of polyethylene superstrate material. (E) SAR values of different superstrate material of proposed antenna. (F) Reflection coefficient of different shaped superstrate of proposed antenna.

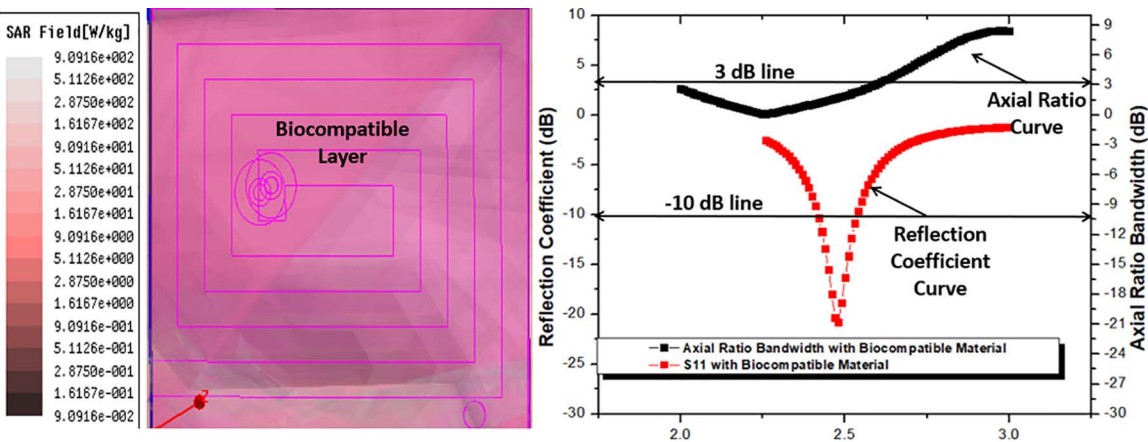

**Fig 7. The SAR, Axial ratio bandwidth, and Reflection coefficient (S₁₁) of proposed antenna with biocompatible layer of Roger rt duroid.**

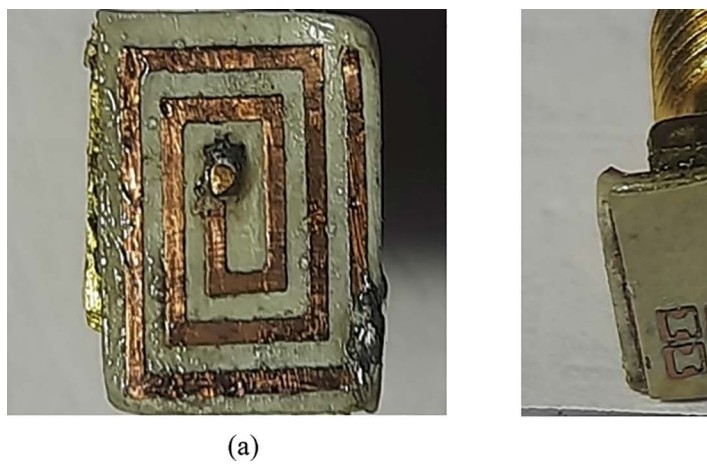
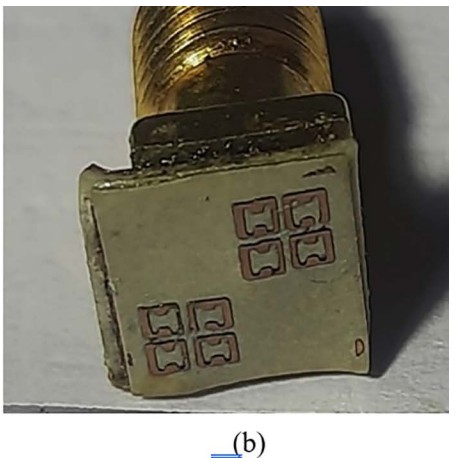

(a)                                                           (b)

**Fig 8. (A) Radiating patch surface of the fabricated implantable antenna.** (B) H-Shaped metamaterial design on the superstrate surface.

and for the testing purpose of the antenna prototype, skin mimicking gel is prepared under specific consideration [23]. For making saline solutions (skin mimicking gel) the amount of sucrose (53%), deionized water (47%), and carbomer (0.5 gram for 40 mL solution) are blended well. This mixture is kept on heating at 80° C for 1 hour. Then it cools down to room temperature. Fig 9A shows the measuring setup and skin-mimicking gel. The suggested implantable antenna's simulated and measured reflection coefficients are thus similar, as shown in Fig 9B.

The fabricated proposed antenna shows a reflection coefficient of -21.67 dB at resonating frequency 2.44 GHz with impedance bandwidth of 90 MHz (2.39 GHz to 2.48 GHz). Slight variation in impedance bandwidth between fabricated and simulated antennas has been observed. This variation is due to not having exact electrical properties of skin mimicking gel as compared to simulation skin model. Superstrate with metamaterial design is situated on patch surface with help of glue. The presence of an air gap in glue may cause some losses too. Overall fabricated and simulated antennas exhibit good agreement in their performance.

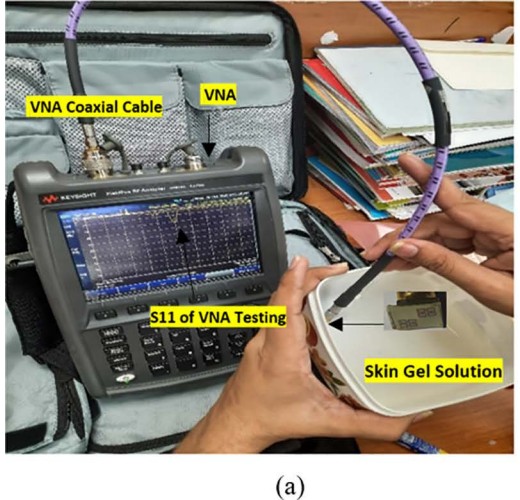 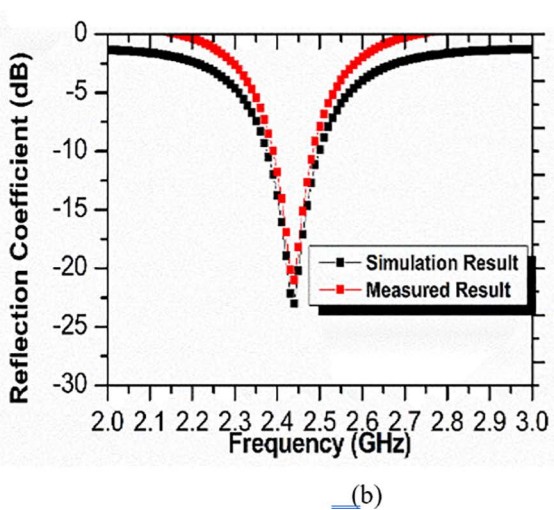

(a)                                                                                          (b)

**Fig 9. (A) Measurement setup of the proposed implantable antenna with saline solution.** (B) Comparison of the reflection coefficient of simulated and fabricated proposed implantable antenna.

## 3.2)   Assessment of SAR

SAR is the most inevitable parameter for implantable antennas to guarantee the protection of people during telemetry sessions. As per IEEE, two standards are given for 1 gram and 10 grams of human tissues. SAR must not be greater than 1.6 W/KG for 1 gram of tissues; else, it will cause the temperature of nearby tissues to rise. This may cause serious health issues for the patients.

For setting input power as 1 watt, the maximum SAR of A/N 1 is 913.3 for 1 gram of tissue. To provide complete safety for the patient, the input power of A/N 1 should be less than 1.75 m watts. For all stepwise designed antennas, SAR values are mentioned in Table 3.

Table 3 makes it very evident that the suggested antenna requires 1.67 mW of input power. This much power is sufficient for biomedical applications. SAR of different designs of implantable antennas is shown in Fig 10.

After comparing the suggested implantable antenna's performance with recently published work, it is strongly advised overall. The proposed antenna exhibits small volume, good CP properties along with an acceptable range of SAR value. The comparison table of recently published work with our proposed implantable antenna is presented in Table 4 below.

## 3.3)   CP performance and gain of proposed implantable antenna

For any microwave antenna to be an implantable antenna, Circular polarization behavior is extremely recommended. During telemetry sessions, the CP behavior of implantable antenna leads to enhancing the patient's mobility and freedom. Proposed antenna performs CP characteristics having an excellent ARBW of 830 MHz in the ISM Band. Concerning axial ratio, CP behavior is well expressed in Fig 4B. The internal mechanism of CP proposed implantable antenna (A/N 3) can

**Table 3. Summary sheet of SAR for different antenna designs with input power for 1 gram of tissue.**

| S. No | Antenna | Resonating Frequency (GHz) | SAR (W/KG) | Input power (mW) |
|-------|---------|----------------------------|------------|------------------|
| 1 | A/N 1 | 2.53 | 913.3 | 1.75 |
| 2 | A/N 2 | 2.46 | 960.5 | 1.66 |
| 3 | A/N 3 | 2.44 | 952.7 | 1.67 |

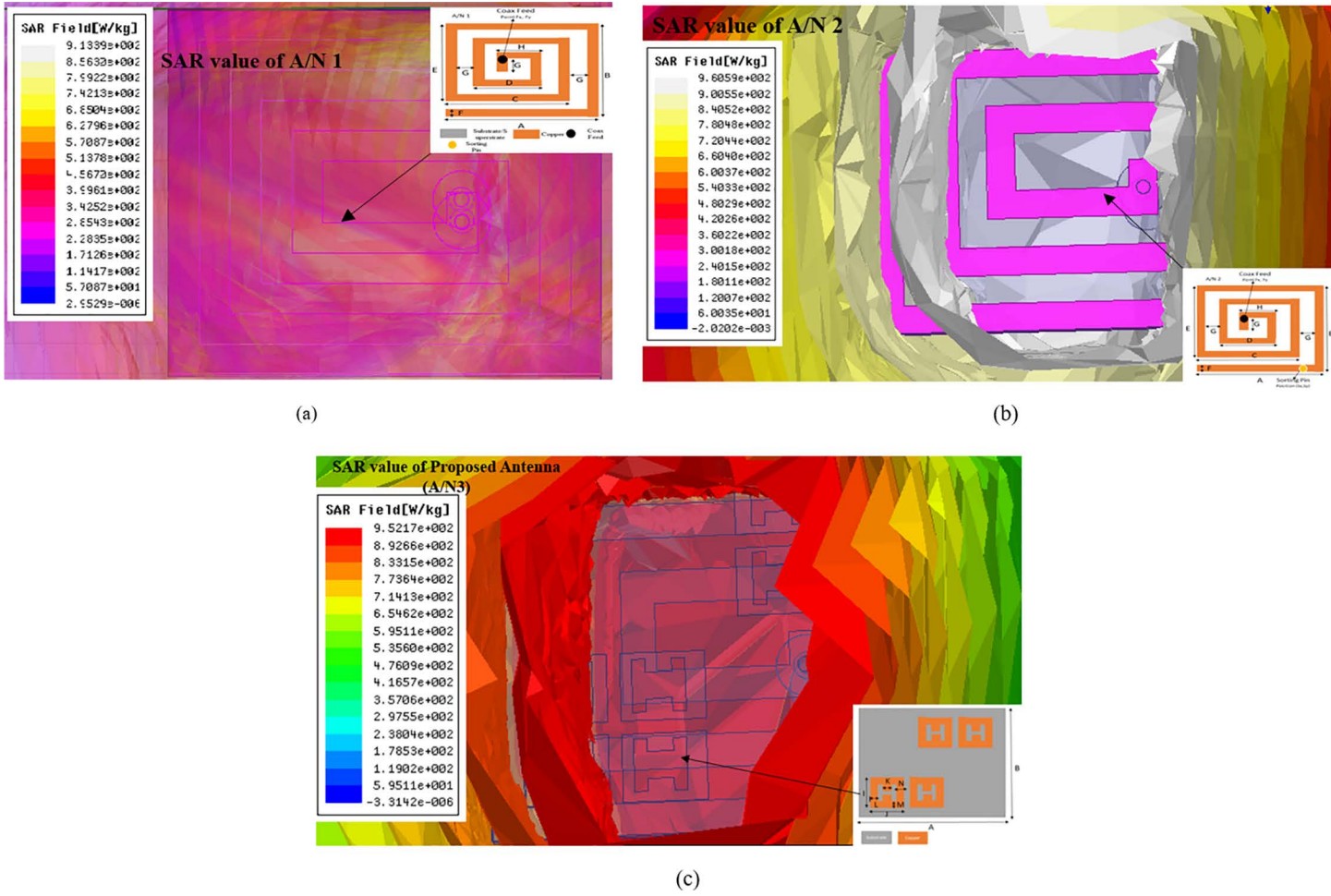

**Fig 10. (A) SAR of A/N 1 according to IEEE standard for 1 gram of tissue.** (B) SAR of A/N 2 according to IEEE standard for 1 gram of tissue. (C) SAR, using an IEEE standard, of the suggested antenna (A/N 3) for one gram of tissue.

be demonstrated by observing surface current distribution. The current distribution on the patch surface of the suggested implantable antenna filled with metamaterial at a resonant frequency of 2.44 GHz is shown in Fig 11. It is clearly shown with four different phases ($0^0$, $90^0$, $180^0$, and $270^0$). From Fig 11 (a), at $0^0$ phase surface current distribution is dominant in the + X direction (rightward). At $90^0$ phase Current distribution is dominant in the -Y direction (downward). The surface current density for $0^0$ and $90^0$ are examined in equal magnitude and just opposite phase with $180^0$, and $270^0$ respectively. This left-hand circular polarization behavior for the metamaterial-loaded suggested implantable antenna (A/N 3) can be explained by the prevailing current rotating clockwise in Fig 11. This Fig 11 of the current distribution, yields the suggested antenna (A/N 3) a perfect circular polarized antenna.

From Fig 12, it is well demonstrated that the proposed antenna (A/N 3) shows the simulated gain of -29.6 dB at 2.44 GHz resonating frequency. From Fig 12C, It is evident that the Left hand circularly polarized (LHCP) gain exceeds to right hand circularly (RHCP) polarized gain by around 28 dB at resonating frequency of 2.44 GHz. Hence this proposed antenna is LHCP and useful to avoid multipath losses for all telemetry sessions. The complete setup of the fabricated antenna is tested for the gain parameter. The measured value of gain is obtained -30 dB at 2.44 GHz. Both simulated

**Table 4. Comparison of recent published work with proposed implantable antenna.**

| Reference | Resonating Frequency (GHz) | Dimension (mm³) | Impedance Bandwidth % | Axial Ratio Bandwidth (MHz) | SAR for 1 gram | CP |
|---|---|---|---|---|---|---|
| [11] | 0.915 | $33.63\lambda_0$ x $33.63\lambda_0$ x $3.88\lambda_0$ | 3.93% | 1.22% | ------ | Yes |
| [12] | 2.45 | $3.14$ x $(39.2\lambda_0)^2$ x $10.12\lambda_0$ | ------ | 18.3% | ------ | Yes |
| [13] | 0.915 | $15.86\lambda_0$ x $17.08\lambda_0$ x $0.7625\lambda_0$ | 18.9% | 17.2% | ------ | Yes |
| [21] | 2.45 | $57.16\lambda_0$ x $49\lambda_0$ x $2\lambda_0$ | 35.8% | 17.14% | 524.3 | Yes |
| [24] | 2.4 | $3.14$ x $(44.92\lambda_0)^2$ x $10.37\lambda_0$ | 8.3% | 2.49% | ------ | Yes |
| **This work** | **2.44** | **$57.16\lambda_0$ x $49\lambda_0$ x $2\lambda_0$** | **4.62%** | **830 MHz or 41.29%** | **952.1** | **Yes** |

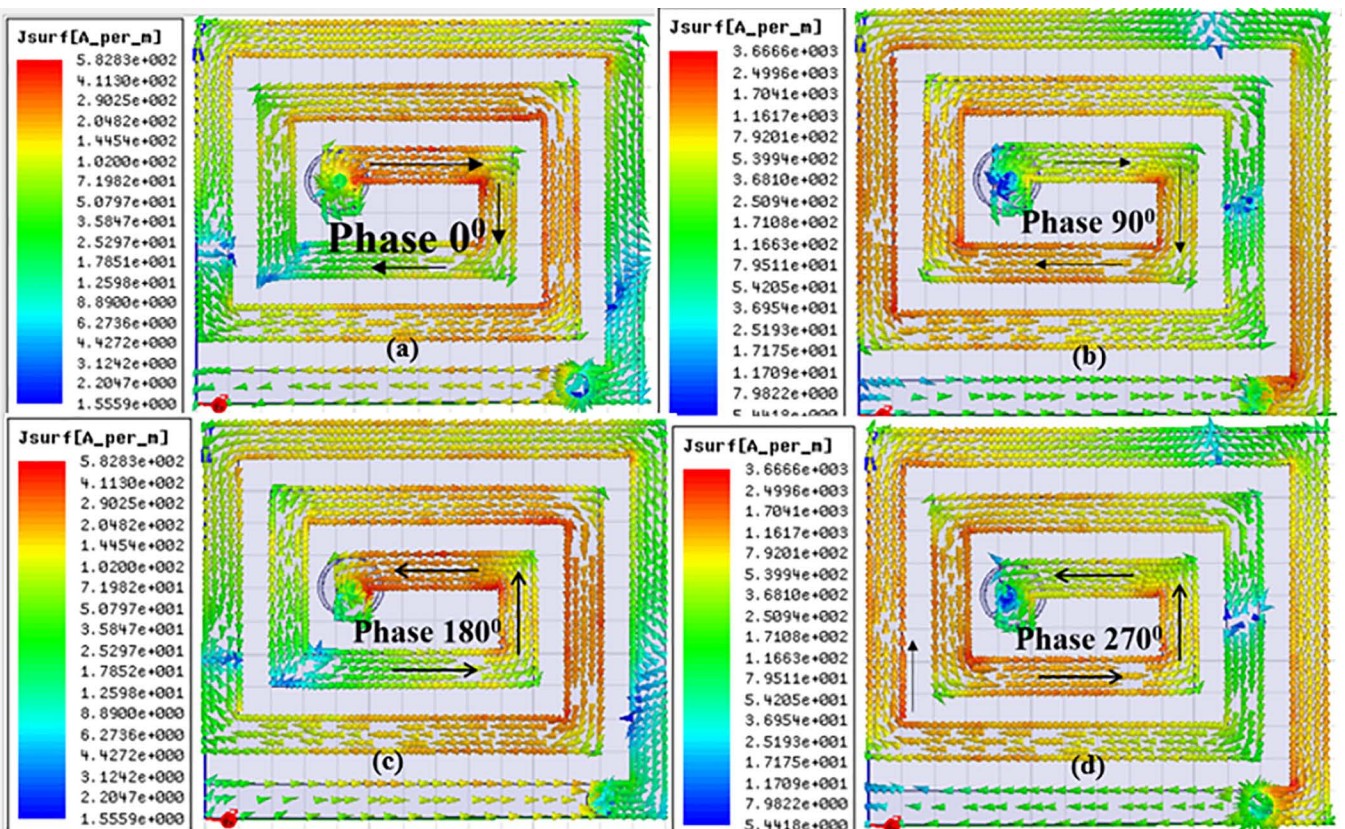

**Fig 11. (A) Surface current distribution of radiating path (patch) proposed implantable antenna with $0^0$ phase angle at 2.44 GHz.** (B) at $90^0$ (C) at $180^0$ (D) at $270^0$.

and measured gain parameters have good agreement. Slight variation may happen due to environmental effects in saline solution.

### 3.4) Analysis of link margin

In this paper, our assumption is to utilize the proposed implantable antenna in the hospital environment for biodata transmission. The telemetry scenario is taken place in a hospital with wireless communication. To substantiate the

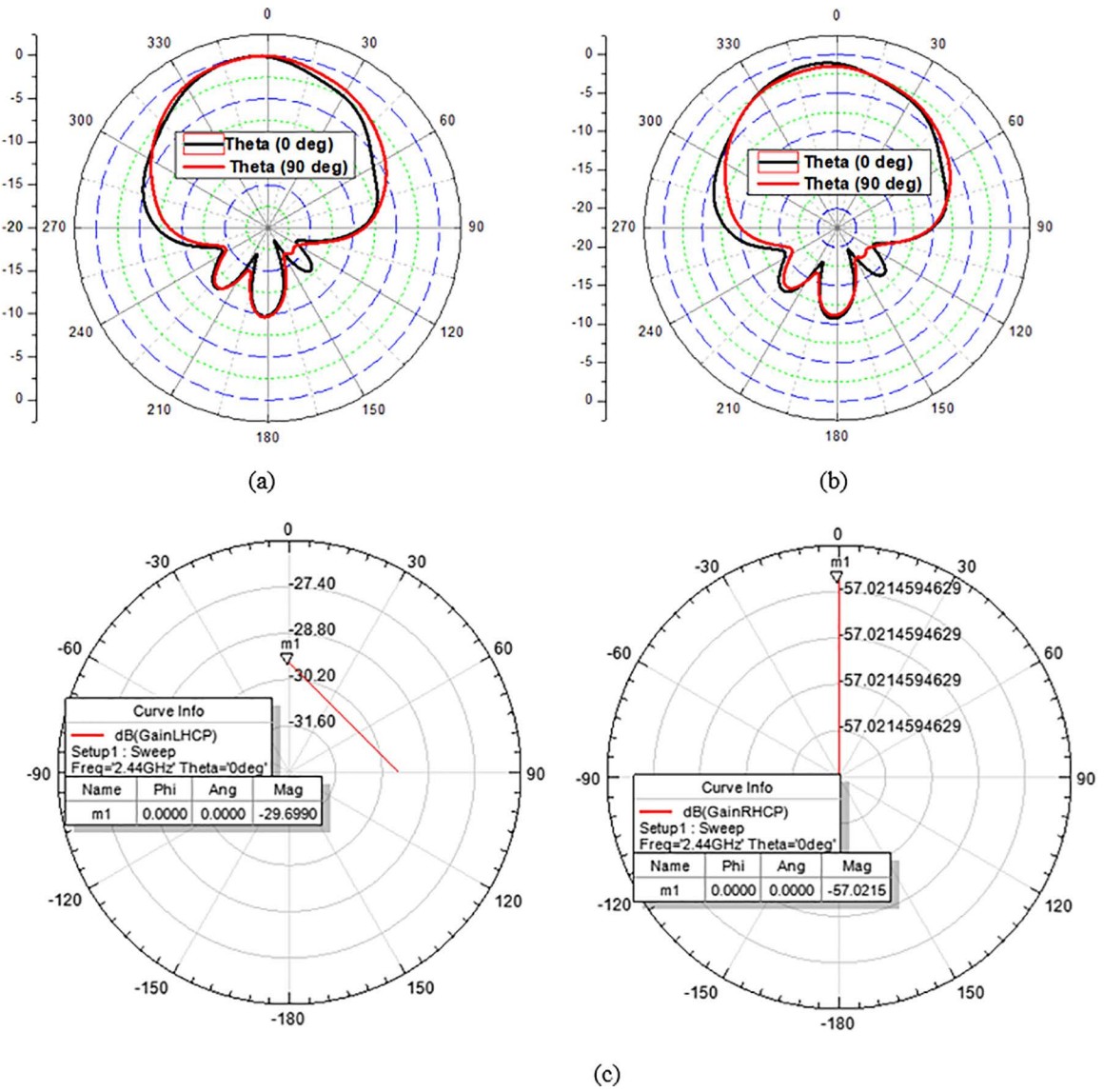

**Fig 12. (A) Normalized simulated gain of suggested antenna at 2.44 GHz.** (B) Normalized measured gain of the proposed antenna at 2.44 GHz. (C) LHCP and RHCP gain of suggested antenna at 2.44 GHz.

communication channel between the external receiver and implanted antenna, the link margin must be at least greater than zero dB. As per the European Research Council Standard [25], we are bounded to use 25 µW transmitted power to this proposed implantable antenna. All the specifications of transmitting antenna, receiver, and signal quality are presented in table 5.

All the mathematical equations for link margin analysis of proposed antenna are discussed in detail [26]. To calculate the link margin, it is essential to find free space loss from the equation given below.

$$L_f\,(dB) = 10\,\log_{10}\,(4\pi d/\lambda)^2 \qquad (1)$$

**Table 5.  Details of Wireless communication parameters.**

| Transmitting End | |
|---|---|
| Resonating Frequency | 2.44 GHz |
| Transmission Power | 25 μW |
| Transmission Power (dBm) | -16 |
| Transmission Antenna Gain | -29.6 dB |
| EIRP | -46.6 dB |
| Feeding Loss | 1 dB |
| **Radiation Propagation** | |
| Distance | Variation |
| Free Space Loss ($L_f$) | Distance dependant |
| **Receiver Station** | |
| Receiving Antenna Gain | 2.15 dB |
| Ambient Temp (K) | 293 |
| Boltzmann Constant | $1.38 \times 10^{-23}$ |
| Noise Power Density ($N_O$) | -199.95 dB |
| Noise Figure | 3.5 |
| **Signal Quality** | |
| Bit Rate $B_r$ | Variation |
| Bit Error Rate | $1 \times 10^{-5}$ |
| $E_b/N_O$ (Ideal PSK) | 9.6 dB |
| Fixing Deterioration | 2.5 dB |

Where d is the distance between the transmitter antenna and receiver antenna.

⅄ is the operating wavelength of the transmitting antenna.

For the different values of d, free space loss is presented in Fig 13A. Link margin calculation is taken for distance d = 4 meter with help of the standard equation given below.

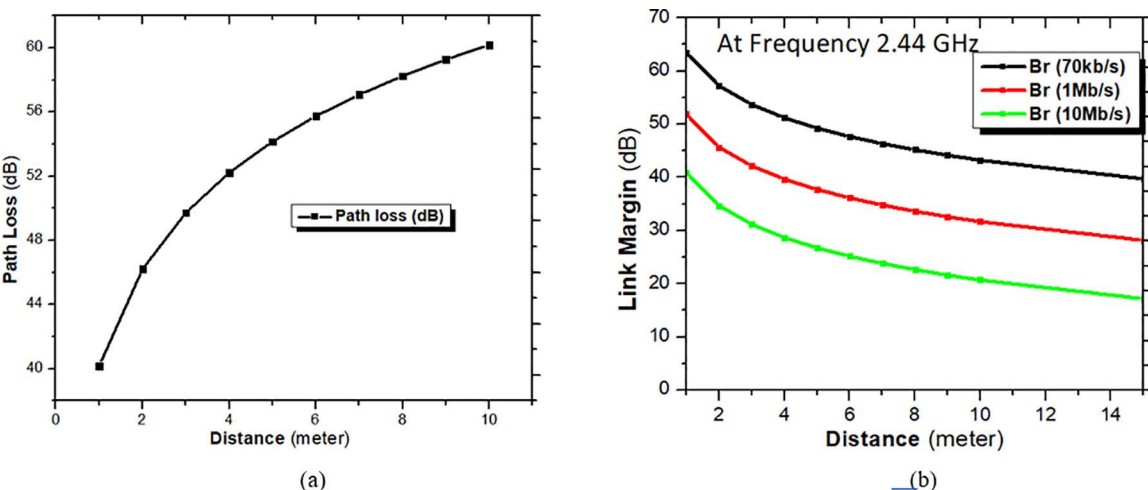

(a)          (b)

**Fig 13.  (A) Path loss of Transmitting antenna at a different distance.** (B) Link margin of the proposed implantable antenna at 2.44 GHz frequency for different bit rates at different distances.

Link Margin= Link C/$N_o$ – Required C/$N_O$

$$\text{Where Link C/}N_O = P_t - L_{feed} + G_t - L_f - L_a + G_r - L_{feed} - N_O \tag{2}$$

$$\text{Required C/}N_o = E_b/N_O + 10 \log \ (B_r) \ -G_c + Gd \tag{3}$$

For distance **d=4-meter, $B_r$= 70 Kb/s**.

$$\text{Link C/}N_o = -46.03 \ -1 - 29.6 - 52.2276 - 0.5 + 2.15 - 1 + 199.95 \tag{4}$$

$$\text{Link C/}N_o = 71.74 \text{dB} \tag{5}$$

$$\text{Required C/}N_o = 9.6 + 48.45 + 2.5 \tag{6}$$

$$\text{Required C/}N_o = 60.55 \text{ dB} \tag{7}$$

Link Margin is calculated below.

$$\text{Link Margin} = 71.74 - 60.55 \tag{8}$$

Link Margin = 11.19 dB

The variation of path loss and link margin w.r.t the distance is explained in Fig 13.

From Fig 13B, it is well demonstrated that a wireless communication link is nicely established between transmitting and receiving antenna with different bit rates at distances from 1 meter to 8 meters. At 4 meters, the link margin of suggested research work is obtained 11.19 dB for a bit rate of 70Kb/s. From link margin analysis of the proposed implantable antenna, it can be said that it is a good candidate to be used for telemetry sessions in biomedical devices for health care monitoring.

## 4. Conclusion

For use in biomedical applications, a highly miniaturized implanted antenna is devised and experimentally validated in saline solution (skin imitating gel). Sorting pin and metamaterial design yield the best results. The proposed antenna's ARBW and impedance bandwidth have precisely covered ISM band (2.4 GHz to 2.48 GHz). In comparison to recently published publications, the proposed antenna (CP) exhibits almost maximum axial ratio bandwidth (830 MHz). The suggested antenna's reduced size (10.27 mm$^3$) results in a respectable SAR 952.1 value. There are no ground slots in the planned study, which could result in false radiation that creates back lobes. In contrast to previous recent research articles, the suggested work is accomplished using straightforward structures and metamaterial approaches. To guarantee full comprehension of future researchers in the same field, every facet of the suggested implanted antenna is covered in detail. With outstanding performance, the important parameters—such as impedance bandwidth, ARBW, size, SAR, straightforward design, CP characteristics, and link margin—are met. The suggested antenna has been successfully constructed and tested. The suggested implanted antenna's measured and simulated outcomes are similar. The suggested implantable antenna has undergone a satisfactory parametric investigation. A comprehensive link margin analysis has guaranteed a strong wireless connection between the sending and receiving antennas. The biomedical applications of the constructed prototype work appear promising.

## Supporting information

**S1 File. Data file of proposed research work.**
(PDF)

## Author contributions

**Conceptualization:** Rajiv Kumar Nehra.

**Data curation:** Rajiv Kumar Nehra.

**Formal analysis:** Rajiv Kumar Nehra, Ajay Dureja.

**Funding acquisition:** Tiansheng Yang, Rajkumar Singh Rathore, Lu Wang.

**Investigation:** Rajiv Kumar Nehra, Ajay Dureja, Tiansheng Yang.

**Methodology:** Rajiv Kumar Nehra, Ajay Dureja.

**Project administration:** Ajay Dureja, Lu Wang.

**Software:** Rajkumar Singh Rathore.

**Supervision:** Tiansheng Yang, Rajkumar Singh Rathore, Lu Wang.

**Validation:** Tiansheng Yang, Rajkumar Singh Rathore, Lu Wang.

**Visualization:** Tiansheng Yang, Rajkumar Singh Rathore, Lu Wang.

**Writing – original draft:** Rajkumar Singh Rathore, Lu Wang.

**Writing – review & editing:** Rajkumar Singh Rathore, Lu Wang.

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
