## [Decision Letter · Decision Letter 0]

10 Feb 2025

PONE-D-24-56406Miniaturised Implantable Circular Polarized Antenna with a High ARBWPLOS ONE

Dear Dr. Rathore,

Thank you for submitting your manuscript to PLOS ONE. After careful consideration, we feel that it has merit but does not fully meet PLOS ONE’s publication criteria as it currently stands. Therefore, we invite you to submit a revised version of the manuscript that addresses the points raised during the review process.

We look forward to receiving your revised manuscript.

Kind regards,

Mustafa Hikmet Bilgehan Ucar, Ph.D.

Academic Editor

PLOS ONE

Journal Requirements:

3. In the online submission form, you indicated that “All data would be available on the specific request to corresponding author.”

Reviewers' comments:

Reviewer's Responses to Questions

**Comments to the Author**

1. Is the manuscript technically sound, and do the data support the conclusions?

Reviewer #1: Partly

Reviewer #2: Partly

2. Has the statistical analysis been performed appropriately and rigorously? 

Reviewer #1: No

Reviewer #2: N/A

3. Have the authors made all data underlying the findings in their manuscript fully available?

Reviewer #1: Yes

Reviewer #2: Yes

4. Is the manuscript presented in an intelligible fashion and written in standard English?

Reviewer #1: Yes

Reviewer #2: Yes

5. Review Comments to the Author

Reviewer #1: The author presented an interesting work on implantable antennas having high axial ratio bandwidth. Although the manuscript has potential, it still needs to be addressed with some concern before making a final decision.

The ISM band ranges from 2400 MHz to 2500 MHz, gloabally. However, the proposed antenna offers a bandwidth of 830 MHz. The author should comment on the improvement of the bandwith using the same antenna.

1: Introduction needs significant improvement by putting more literature work and propoerly citing the claims. For instance, the statement "The implantable antenna is a necessary part of biomedical equipment in order to establish wireless communication between external receivers and Implantable Medical devices (IMDs)." must be cited by 10.34133/cbsystems.0172. Similarly, "Antenna size, radiation efficiency, low SAR, patient safety, biocompatibility, and

circular polarization are highly concerned parameters to developing quality IMDs in the medical health system" can be cited by 10.1088/1402-4896 || 10.1016/j.heliyon.2024.e40627. Moreover, " However, the antenna's

performance suffers due to its significant size decrease" can be cited by 10.1109/LAWP.2024.3398076 and 10.3390/mi14101842.

2: The state of the work must be added in literature review and discussed in such a way that the proposed work by author show its major contribution. 10.1109/TAP.2024.3454434 | 10.1109/TAP.2024.3499367 | 10.1109/TAP.2024.3503919

3: The presentation of Fig. 1 can be updated for better understanding; please refer to the 10.1109/JSEN.2024.3423023

4: The size of Figure. 2 must be resized; display it properly side by side.

5: More dimensions of the antenna must be added in Fig. 2 (a) and Fig. 2(c).

6: Design procedure need more explanation by explaining them mathematically as explained in 10.3390/s23020709 || 10.1109/TWC.2024.3480353

7: In Fig. 3(c), how did the researchers come up with the 4-element design? Why is not the full metameterial mesh is utilized? The author must explain it in detail, as the idea looks very strange with reference to state-of-the-art work realted to metamaterial.

8:Fig. 4(e) must be explained clearly and in more detail.

9: What is the purpose of Fig. 5(a)? since the parameter has no effect on the performance of the antenna.

10: Section 2.2 b) must add discussion about the 4 layered human tissue model as utilized in 10.1016/j.rineng.2024.103818 || 10.46620/URSIATRASC24/RBLP2414

11: Figure 6 is so confusing, thus the author should replot them by adding clear pictures that are easy to understand.

12: In Fig. 9, a clear picture of the VNA must be added to verify the s-parameter findings.

13: The SAR value is too high for an implantable antenna, even with a low power of 1 mW. What is the power limit to bing the SAR in an acceptable range?

14: It is really difficult to extract knowledge from Fig. 10. The author should replot the figure by plotting the SAR graph in a transparent manner.

15: Figure 11 is not clear; it is hard to understand the CP mechanism. The author must add a high-quality picture through which the reader can understand the CP mechanism.

16: Figure 12 should be replotted using professional tools and a high-quality image.

17: A number of uncited equations are used in Section 3.4, the author must cite proper reference from where the equations are extracted. The following reference can be utilized for the said purpose: Intelligent metasurface-based antenna with pattern and beam reconfigurability for internet of things applications, AEJ, 2024.

18: Figure 13 should be replaced by a high-quality image.

19: Conclusion must be revised by adding more details and putting major focus on the contribution of the antenna.

20: More literature must be added by putting references from the journal to relate your work with the scope of the journal. Here are some suggestions 10.1371/journal.pone.0276922 || 10.1371/journal.pone.0306446 || 10.1371/journal.pone.0301924

Reviewer #2: Review of the article "Miniaturised Implantable Circular Polarized Antenna with a High

ARBW" submitted to the PLOS ONE

1. The article aligns well with the journal's scope and is presented in an organized manner,

making it easy to read and follow.

2. The authors propose a design for an implanted antenna that incorporates metamaterial

and a sorting pin for biological applications. This antenna operates at a frequency of 2.44

GHz within the ISM band.

3. A highly miniaturized implanted antenna is devised and experimentally validated in

saline solution (skin-imitating gel) for use in biomedical applications.

4. The authors have compared the proposed antenna's performance with those recently

published in the literature to reveal its robustness.

5. The methodology is clearly presented, with extensive explanations and discussions of

both the method and the results.

However, I suggest the authors address the following issues:

1. Return loss in dB is positive, and the reflection coefficient is negative. Please take care

of the terms and their physical meaning. The term (Return loss) should be replaced,

wherever it is mentioned in the text, by either (input reflection coefficient) or only (S11).

For more information, the authors are advised to see:

T. S. Bird, "Definition and Misuse of Return Loss [Report of the Transactions Editor-inChief]," in IEEE Antennas and Propagation Magazine, vol. 51, no. 2, pp. 166-167, April

2009.

doi: 10.1109/MAP.2009.5162049

2. In Table 4, expressing the dimensions of the presented antennas in terms of the design

wavelengths rather than absolute units would strengthen the comparison. However, I

suggest the authors keep only the antennas with circular polarization.

3. Subheadings directly after headings, as in Section 3, are not preferred. Please add one or

two sentences to introduce the section and keep the paper flowing.

4. I suggest the authors slightly revise the language of the article. The attached file includes

many suggestions.

6. PLOS authors have the option to publish the peer review history of their article (what does this mean? ). If published, this will include your full peer review and any attached files.

**Do you want your identity to be public for this peer review?** For information about this choice, including consent withdrawal, please see our Privacy Policy .

Reviewer #1: No

Reviewer #2: **Yes: ** Prof. Jawad K. Ali

---

## [Author Response · Author response to Decision Letter 1]

18 Feb 2025

I am very grateful for the time you took out for my research paper.

Reviewer 1

Comment Response

The author presented an interesting work on implantable antennas having high axial ratio bandwidth. Although the manuscript has potential, it still needs to be addressed with some concern before making a final decision.

The ISM band ranges from 2400 MHz to 2500 MHz, globally. However, the proposed antenna offers a bandwidth of 830 MHz The author should comment on the improvement of the bandwidth using the same antenna. We are highly obliged to have your appreciation words for our research work. Thank you and Regards. The proposed antenna offers huge axial ratio bandwidth which is useful to create multiple channels with high bit rate data transmission and reduce multipath fading losses up to large extents. The high bandwidth ensures patient safety during data telemetry transmission.

1. Introduction needs significant improvement by putting more literature work and properly citing the claims. For instance, the statement "The implantable antenna is a necessary part of biomedical equipment in order to establish wireless communication between external receivers and Implantable Medical devices (IMDs)." must be cited by 10.34133/cbsystems.0172. Similarly, "Antenna size, radiation efficiency, low SAR, patient safety, biocompatibility, and

circular polarization are highly concerned parameters to developing quality IMDs in the medical health system" can be cited by 10.1088/1402-4896 || 10.1016/j.heliyon.2024.e40627. Moreover, " However, the antenna's performance suffers due to its significant size decrease" can be cited by 10.1109/LAWP.2024.3398076 and 10.3390/mi14101842.

As suggested’ The literature work has been added in introduction part of manuscript. All the recommended citation of related research work has been added and also expressed the research gaps of these publication. In comparison to each recommended citation, authors have emphasised their novel research work and performance. The appropriate changes have been added to page number 1 with yellow highlights.

Thank you and regards

2. The state of the work must be added in literature review and discussed in such a way that the proposed work by author show its major contribution. 10.1109/TAP.2024.3454434 | 10.1109/TAP.2024.3499367 | 10.1109/TAP.2024.3503919.

As suggested’ the recommended citations are added in introduction part of revised manuscript. The novelty of our research work is also expressed in comparison of all suggested citations. The appropriate changes are highlighted at Page No 2. Thank you and regards

3. The presentation of Fig. 1 can be updated for better understanding; please refer to the 10.1109/JSEN.2024.3423023

As per received suggestions, the presentation of Fig 1 is updated in revised manuscript at page no 3. Thank you and regards

4. The size of Figure. 2 must be resized; display it properly side by side.

As per received suggestions, existing images in this Fig are replaced by the high-quality tiff file images, resized, and arranged side by side in manuscript at page no 4. Thank and regards

5. More dimensions of the antenna must be added in Fig. 2 (a) and Fig. 2(c). As per received suggestions, the Fig 2 (a) and Fig 2(c) are presented with more dimensions and high quality. Thank and regards

6. Design procedure need more explanation by explaining them mathematically as explained in 10.3390/s23020709 || 10.1109/TWC.2024.3480353 As suggested’ the design procedure is explained better from recommended citations work.

Thank and regards

7. In Fig. 3(c), how did the researchers come up with the 4-element design? Why is not the full metamaterial mesh is utilized? The author must explain it in detail, as the idea looks very strange with reference to state-of-the-art work related to metamaterial.

In our research work, we have used metamaterial superstrate in order to improve radiation properties of antenna. The 2 x 2 MMT array (4 elements) has been used for improved radiation and axial ratio bandwidth. The major concern for this research work is to have high broadband axial ratio bandwidth and better radiation properties. The motivation of using this array size is taken from [21]. Authors have used only four elements MMT designs to meet broad band CP characteristics and fulfilled the research gap of past years. Authors have obtained huge axial ratio bandwidth with simple design of low order of MMT array. By considering simple design fabrication and broad bandwidth, authors have used only 4 elements. The array size undoubtedly can be increased on superstrate surface. This scope of improvement related to directive gain and SAR, will be carried forward in the future extension of this research work. Thank you and regards

8. Fig. 4(e) must be explained clearly and in more detail. As suggested’ The Fig. 4 (e) is explained in detail at Page No. 8. Thank you and regards

9. What is the purpose of Fig. 5(a)? since the parameter has no effect on the performance of the antenna.

As suggested’ The Fig. 5(a) shows the resilience capability of proposed antenna. This Fig demonstrates the credible performance of proposed antenna irrespective of penetration depth inside human body tissue. It means that the proposed antenna performance is not altered due to positional change in human body environment. Thank you and regards

10. Section 2.2 b) must add discussion about the 4 layered human tissue model as utilized in 10.1016/j.rineng.2024.103818 || 10.46620/URSIATRASC24/RBLP2414

Respected Sir/Mam,

The complete research is simulated and tested under vitro model analogy where only single human body tissue is utilised to validate the antenna performance. All four body tissues like skin, fat, bone, muscle are examined separately. The proposed antenna is optimised well in ISM band only in skin tissues. Thank you and regards

11. Figure 6 is so confusing; thus the author should replot them by adding clear pictures that are easy to understand.

In Fig. 6, The authors have investigated detailed parametric study on reflection coefficient, SAR based on superstrate material and shape selection. The Fig. 6 reveals that proposed antenna is highly influenced by material and shape of superstrate. Based on this parametric study, authors have finalised the Superstrate material (Rogers rt Duroid) and structure (rectangular) which ensures performance in ISM bands. As suggested’ the Fig. 6 is updated with clear pictures. Thank you and regards

12. In Fig. 9, a clear picture of the VNA must be added to verify the s-parameter findings. As suggested’ clear picture of VNA is added in manuscript at page no 13.

Thank you and regards

13. The SAR value is too high for an implantable antenna, even with a low power of 1 mW. What is the power limit to bing the SAR in an acceptable range?

Respected Sir/Mam,

As per IEEE standard 1999, 2005, the SAR value for 1 gram and 10 gram of body tissue must not exceed 1.6 W/KG and 2 W/KG respectively. It means that for 1gram and 1 mw input power, the SAR value should not exceed 1600 w/kg. In our research work, the SAR of proposed design is obtained 952.7 which is quite safe. From this SAR value, antenna can be fed with maximum 1.67 mW input power. Thank you and regards

14. It is really difficult to extract knowledge from Fig. 10. The author should replot the figure by plotting the SAR graph in a transparent manner.

As per received suggestion, authors have used clear and transparent plots of SAR graph presented well at page no 14. Thank you and regards

15. Figure 11 is not clear; it is hard to understand the CP mechanism. The author must add a high-quality picture through which the reader can understand the CP mechanism. As per received suggestion, Authors have added high quality picture of CP mechanism presented at page no 16. Thank you and regards

16. Figure 12 should be replotted using professional tools and a high-quality image. As per received suggestion, the images have been used with high quality presented at page no 16.

Thank you and regards

17. A number of uncited equations are used in Section 3.4, the author must cite proper reference from where the equations are extracted. The following reference can be utilized for the said purpose: Intelligent metasurface-based antenna with pattern and beam reconfigurability for internet of things applications, AEJ, 2024.

As per received suggestion, the recommended publication has been cited in manuscript at page no 18. Thank you and regards

18. Figure 13 should be replaced by a high-quality image. As suggested’ authors have replaced Fig 13 with new high-quality images presented at page no 18. Thank you and regards

19. Conclusion must be revised by adding more details and putting major focus on the contribution of the antenna. As per received suggestions, authors have added more details and put complete focus on contribution of proposed antenna at page no 19. Thank you and regards

20. More literature must be added by putting references from the journal to relate your work with the scope of the journal. Here are some suggestions 10.1371/journal.pone.0276922 || 10.1371/journal.pone.0306446 || 10.1371/journal.pone.0301924

As per received suggestions, authors have added all recommended citations which are present at page no 2. Thank you and regards

Reviewer 2

1. The article aligns well with the journal's scope and is presented in an organized manner,

making it easy to read and follow.

Respected Sir/Mam,

Thank you for your kind words

Regards

2. The authors propose a design for an implanted antenna that incorporates metamaterial

and a sorting pin for biological applications. This antenna operates at a frequency of 2.44

GHz within the ISM band.

Respected Sir/Mam,

Thank you for your kind words

Regards

3. A highly miniaturized implanted antenna is devised and experimentally validated in

saline solution (skin-imitating gel) for use in biomedical applications.

Respected Sir/Mam,

Thank you for your kind words

Regards

4. The authors have compared the proposed antenna's performance with those recently

published in the literature to reveal its robustness.

Respected Sir/Mam,

Thank you for your kind words

Regards

5. The methodology is clearly presented, with extensive explanations and discussions of

both the method and the results.

Respected Sir/Mam,

Thank you for your kind words Regards

Reviewer Comments Authors Response

1. Return loss in dB is positive, and the reflection coefficient is negative. Please take care

of the terms and their physical meaning. The term (Return loss) should be replaced,

wherever it is mentioned in the text, by either (input reflection coefficient) or only (S11).

For more information, the authors are advised to see:

T. S. Bird, "Definition and Misuse of Return Loss [Report of the Transactions Editor-in Chief]," in IEEE Antennas and Propagation Magazine, vol. 51, no. 2, pp. 166-167, April

2009.

doi: 10.1109/MAP.2009.5162049

As per received suggestions, authors have read recommended article in detail and consequently the term Return loss is replaced with input reflection coefficient in the entire manuscript.

Thank you and regards

2. In Table 4, expressing the dimensions of the presented antennas in terms of the design

wavelengths rather than absolute units would strengthen the comparison. However, I

suggest the authors keep only the antennas with circular polarization.

As per received suggestions, In Table 4, authors have expressed antennas dimension in term of wavelength and used all CP antennas for fair comparison.

Thank you and regards

3. Subheadings directly after headings, as in Section 3, are not preferred. Please add one or

two sentences to introduce the section and keep the paper flowing.

As per received suggestions, authors have mentioned appropriate sentences just after main heading of section 3 at page no 12.

Thank you and regards

4. I suggest the authors slightly revise the language of the article. The attached file includes

many suggestions.

As per received suggestions, authors have wisely read the attached file. All suggestions are carefully considered and updated in the revised manuscript.

Thank you and regards

---

## [Decision Letter · Decision Letter 1]

11 Mar 2025

Miniaturised Implantable Circular Polarized Antenna with a High ARBW

PONE-D-24-56406R1

Dear Dr. Rathore,

We’re pleased to inform you that your manuscript has been judged scientifically suitable for publication and will be formally accepted for publication once it meets all outstanding technical requirements.

Kind regards,

Mustafa Hikmet Bilgehan Ucar, Ph.D.

Academic Editor

PLOS ONE

Additional Editor Comments (optional):

Reviewers' comments:

Reviewer's Responses to Questions

**Comments to the Author**

1. If the authors have adequately addressed your comments raised in a previous round of review and you feel that this manuscript is now acceptable for publication, you may indicate that here to bypass the “Comments to the Author” section, enter your conflict of interest statement in the “Confidential to Editor” section, and submit your "Accept" recommendation.

Reviewer #1: All comments have been addressed

Reviewer #2: All comments have been addressed

2. Is the manuscript technically sound, and do the data support the conclusions?

Reviewer #1: Yes

Reviewer #2: Yes

3. Has the statistical analysis been performed appropriately and rigorously? 

Reviewer #1: Yes

Reviewer #2: N/A

4. Have the authors made all data underlying the findings in their manuscript fully available?

Reviewer #1: Yes

Reviewer #2: Yes

5. Is the manuscript presented in an intelligible fashion and written in standard English?

Reviewer #1: Yes

Reviewer #2: Yes

6. Review Comments to the Author

Reviewer #1: The authors carefully address the comments, based upon their response the manuscript is recommended for publication.

Reviewer #2: The authors addressed the reviewers' feedback and resolved their concerns, resulting in a significantly improved revised paper compared to the original submission.

7. PLOS authors have the option to publish the peer review history of their article (what does this mean? ). If published, this will include your full peer review and any attached files.

**Do you want your identity to be public for this peer review?** For information about this choice, including consent withdrawal, please see our Privacy Policy .

Reviewer #1: No

Reviewer #2: **Yes: ** Jawad K. Ali

---

## [Editor Report · Acceptance letter]

PONE-D-24-56406R1

PLOS ONE

Dear Dr. Rathore,

I'm pleased to inform you that your manuscript has been deemed suitable for publication in PLOS ONE. Congratulations! Your manuscript is now being handed over to our production team.

Kind regards,

on behalf of

Dr. Mustafa Hikmet Bilgehan Ucar

Academic Editor

PLOS ONE